# Overrepresentation of New Workers in Jobs with Multiple Carcinogen Exposures in Canada

**DOI:** 10.3390/ijerph21081013

**Published:** 2024-08-01

**Authors:** Disann Katende, Elizabeth Rydz, Emma K. Quinn, Emily Heer, Raissa Shrestha, Sajjad S. Fazel, Cheryl E. Peters

**Affiliations:** 1CAREX Canada, School of Population and Public Health, University of British Columbia, Vancouver, BC V6T 1Z3, Canada; disann.katende@ubc.ca (D.K.); equinn@carexcanada.ca (E.K.Q.);; 2Department of Oncology, Cumming School of Medicine, University of Calgary, Calgary, AB T2N 1N4, Canada; 3BC Centre for Disease Control, Vancouver, BC V5Z 4R4, Canada; 4BC Cancer, Vancouver, BC V5Z 4E6, Canada

**Keywords:** young workers, new workers, occupational hazards, carcinogen, occupational safety

## Abstract

**Background.** In Canada, understanding the demographic and job-related factors influencing the prevalence of new workers and their exposure to potential carcinogens is crucial for improving workplace safety and guiding policy interventions. **Methods.** Logistic regression was performed on the 2017 Labour Force Survey (LFS), to estimate the likelihood of being a new worker based on age, industry, occupation, season, and immigration status. Participants were categorized by sector and occupation using the North American Industry Classification System (NAICS) 2017 Version 1.0 and National Occupational Classification (NOC) system 2016 Version 1.0. Finally, an exposures-per-worker metric was used to highlight the hazardous exposures new workers encounter in their jobs and industries. **Results.** Individuals younger than 25 years had 3.24 times the odds of being new workers compared to those in the 25–39 age group (adjusted odds ratios (OR) = 3.24, 95% confidence interval (95% CI) = 3.18, 3.31). Recent immigrants (less than 10 years in the country) were more likely to be new workers than those with Canadian citizenship (OR 1.36, 95% CI: 1.32, 1.41). The total workforce exposures-per-worker metric using CAREX Canada data was 0.56. By occupation, new workers were the most overrepresented in jobs in natural resources and agriculture (20.5% new workers), where they also experienced a high exposures-per-worker metric (1.57). **Conclusions.** Younger workers (under 25 years) and recent immigrants who had arrived 10 or fewer years prior were more likely to be new workers, and were overrepresented in jobs with more frequent hazardous exposures (Construction, Agriculture, and Trades).

## 1. Introduction

Almost half of the known or suspected carcinogens defined by the International Agency for Research on Cancer can be found in the workplace and often at exposure levels exceeding those found in the general environment [1]. The hazardous effects of these substances underscore the importance of identifying whether certain workers at the intersections of marginalization could be at disproportionate risk of occupational injury and disease. One group of interest is new workers, generally referring to people of any age that are new to a particular workplace, or facing new hazards in their job for any reason [2].

While the definition of a new worker is not consistent [3], for the present investigation, we define new workers as those who have been on the job for less than 6 months [4]. It is well established that new workers are at a higher risk of occupational injury than established workers, and this risk decreases on a continuum the longer that a worker has been in a current job [4]. This elevated risk of injury and illness at work may be due to inadequate training on occupational hazards, failure to communicate concerns, and uncertainty about workplace expectations [5,6,7]. Additionally, new workers are also more likely to be involved with precarious employment, including temporary work, gig employment, and working for cash outside of a standard work relationship [8,9,10]. Precarious employment is generally less stable and provides fewer workplace protections for employees, which may increase the risk of exposure to hazards, including carcinogens [11]. In Canada and other developed nations, occupational exposure is a significant determinant of cancer incidence [12,13]. In 2011, approximately 4% of all newly diagnosed cancer cases were attributed to workplace-related exposures [12]. This highlights the critical need for stringent occupational health and safety regulations to mitigate cancer risks among workers.

Despite these concerns, data on occupational exposure to carcinogens are rarely (if ever) collected by basic demographics like sex or age, let alone job tenure, that would allow conclusions to be drawn on new workers’ hazardous exposures. In this study, we aim to describe the demographic and job-related predictors of being a new worker (job tenure of less than 6 months) in Canada and descriptively examine carcinogen exposures in jobs and industries where new workers are overrepresented.

## 2. Methods

### 2.1. Study Design and Data Collection

We used two main data sources for our analyses, the Canadian Labour Force Survey (LFS) and carcinogen exposure information from CAREX Canada. The LFS is a national cross-sectional survey that aims to assess labour market conditions and to estimate employment and unemployment rates at the national, provincial, territorial, and regional levels [14,15]. In brief, LFS interviews were conducted monthly by telephone interviewers via computer-assisted telephone interviews or by personal visits from a field interviewer [15]. The sampled households are representative of the civilian, non-institutionalized population 15 years of age or older [14]. Those excluded from the survey are persons living on a reserve and other Indigenous settlements, full-time members of the Canadian Armed Forces, the institutionalized population, and households in extremely remote areas with very low population density (an estimated 2% of the population aged 15 and older) [14].

The LFS categorizes workers by sector and occupation using the North American Industry Classification System (NAICS) 2017 Version 1.0 and the National Occupational Classification (NOC) system 2016 Version 1.0 [15,16,17]. Additionally, the survey collects detailed information on personal characteristics of the working-age population, including age, sex, marital status, educational attainment, and family characteristics. In 2017, the LFS further expanded its scope and included a question on the immigration status of participants [14]. For the present study, all monthly surveys from 2017 were pooled to examine the effect of seasonality and month of the survey, which is important for contextualizing seasonal work and when considering new workers.

New worker status was derived from the job tenure with the current employer (in months). Employees who had worked for less than six months were considered new. Other variables of interest included age (categorized into four groups: <25 years old, 25–39, 40–59, and 60+ years old), season (derived from month of survey, into categories of summer (June-August), fall (September-November), winter (December-February). and spring (March to May)), and immigration status (citizen, recent (<10 years since arrival) immigrant, and former (10+ years since arrival) immigrant).

Since there are no data on occupational exposures for new workers measured in workplaces in Canada to quantitatively examine differences in exposure levels by job tenure, we used carcinogen exposure information from CAREX Canada to highlight the potential for exposures among new workers. CAREX Canada is a national workplace exposure surveillance program that provides estimates of the prevalence of occupational exposure to a suite of approximately 50 known and suspected carcinogens [1]. We tabulated exposure information from CAREX Canada alongside the industries and occupations where new workers are overrepresented (i.e., they represent more than the national average proportion of new workers in the labour force of Canada).

### 2.2. Analysis

Bivariate relationships between the outcome and predictor variables in the LFS were performed using Pearson’s Chi-squared test. The significance level was set at α = 0.05, and the p-value was adjusted using the Bonferroni correction for multiple testing. The likelihood of being a new worker was derived with multivariate logistic regression using maximum likelihood (ML) estimation. Four predictors were included in the occupation model: age group, immigration status, occupation, and season. Industry was presented in a separate model from occupation due to the high degree of correlation between these two factors (Appendix A).

Analyses of the LFS were conducted using the R Statistical package (version 4.2.2; R Core Team, 2022) on Windows 10 x64 (build 22621). Study-specific ethics approval was exempted for this study by the Research Ethics Board of the University of British Columbia, as all data were in the public domain [18]. While the LFS is designed to be representative of the Canadian population and the public use microdata files include weights to allow projection to the full working population, we pooled the study participants from each monthly LFS cycle in 2017 to account for seasonal trends in the year, and thus did not apply the developed weights to the calculations.

To descriptively assess the potential risks associated with certain occupations where new workers are overrepresented, we used an exposures-per-worker metric based on CAREX Canada’s estimates of the prevalence of exposure to occupational carcinogens by industry and occupation [1,19]. To calculate the exposures-per-worker metric, the total number of workers exposed to various carcinogenic substances in a specific occupation or industry were summed and divided by the total number of workers in that occupation or industry. The industries and occupations with disproportionately higher proportions of new workers were tabulated alongside their exposures-per-worker metrics, along with a listing of the most prevalent carcinogenic exposures in those settings.

## 3. Results

The 2017 monthly Labour Force Surveys included 608,724 participants (response rate of 82%). Of these, 66,775 individuals (11.0%) were identified as new workers (Table 1). Younger workers were much more likely to be new on the job than older workers, with 41.2% of workers under 25 years being new, compared to only 3.2% of workers aged 60 and over. Workers with lower levels of education were more likely to be new workers, as were recent immigrants and workers in certain industries, including accommodation and food services, and retail trade. One-third of established workers were union members, compared to just 14.9% of new workers.

New workers were more likely to be employed by smaller firms (28%), which is in contrast to only 17.8% of established workers. Among new workers, 60% had permanent jobs, as compared to almost 90% of established workers with permanent jobs. New workers also had lower salaries than more established workers (Table 1). As expected, a seasonal effect was noted, where more survey respondents reported being new workers in the summer months compared to other times of the year. Finally, although the distribution of workers was similar across all regions, new workers were slightly overrepresented in the Atlantic Provinces and British Columbia, compared to the overall workforces in those regions.

Figure 1 summarizes the odds ratios (ORs) of new worker status for both the occupation and industry models. Statistically significant differences in age, sex, educational attainment, immigration status, occupation, and industry were detected between new workers and established workers. Detailed results from the adjusted logistic regression analyses are presented in Appendix A. Survey participants under the age of 25 were more than three times as likely to be new workers compared to those in the 25–39 age group (OR = 3.24, 95% CI: 3.18, 3.31). Recent immigrants (less than 10 years in the country) were more likely to be new workers than those with Canadian citizenship (OR 1.36, 95% CI: 1.32, 1.41). In contrast, there was no difference between immigrants that arrived at least 10 years ago and those with citizenship (OR 0.97; 95% CI: 0.94, 1.00). Survey respondents were more likely to report being a new worker during the summer season (OR 1.45, 95% CI: 1.41, 1.48) when compared to spring.

Compared to the management occupations, all other occupations had an overrepresentation of new workers. However, the most overrepresentation was observed in natural resources and agriculture (OR 3.99, 95% CI: 3.73, 4.27), trades, transportation and equipment operators (OR 2.78, 95% CI: 2.63, 2.95), and art, culture, recreation, and sport (OR 2.58, 95% CI: 2.39, 2.78). Compared to public administration (i.e., governments), the following industries had the most overrepresentation of new workers: business and building services (OR 2.60, 95% CI: 2.45, 2.75), agriculture, forestry, fishing, and hunting (OR 2.51, 95% CI: 2.34, 2.68), and construction (OR 2.43, 95% CI: 2.32, 2.56).

Table 2 and Table 3 are shown below. For the results by industry (Table 2), the overall workforce comprised 11% new workers, with an exposures-per-worker metric of 0.56. New workers comprised 20% of the workforce in accommodation and food services with an exposures-per-worker metric of 0.44, which is relatively low compared to other industries. However, new workers were overrepresented in agriculture, forestry, fishing, and hunting (18.4% of sector), as well as construction (16.6% of sector), with exposures-per-worker metrics of 1.29 and 1.34, respectively.

The most common exposures in these industries include solar radiation, pesticides, wood dust, and silica. For occupational groups (Table 3), new workers were most overrepresented in natural resources and agricultural jobs (20.5% of that job title), where they experienced an exposures-per-worker metric of 1.57. They were also overrepresented in occupational groups with relatively low exposures-per-worker, such as art, cultural, recreation, and sport jobs and in sales and service. In line with the results by industry, new workers were overrepresented in trades, transport, and equipment operation jobs (13.6% of that job title), where they experienced an exposures-per-worker metric of 1.98.

## 4. Discussion

We sought to examine the new worker population in Canada and situate it in the context of their potential for higher risk of exposure to hazardous substances (known and suspected carcinogens) than established workers in Canada. We found that new workers were more likely to be younger, recent immigrants, earning lower salaries, hired in the summer, holding temporary positions, having lower educational attainment, and working in small-sized firms. We also demonstrated that new workers are overrepresented in certain sectors and occupations with a higher likelihood of exposure to known and suspected carcinogens, most concerning in agricultural and construction settings.

While we identified other sectors and occupations where new workers were overrepresented but had a lower likelihood of multiple carcinogen exposures (e.g., sales and service sector jobs), this does not imply that these workers are risk-free. It is important to note that CAREX Canada does not document all potential chemical, physical, and dust exposures that may occur in these settings. Consequently, workers in these sectors may still face significant exposure to other hazardous substances not currently captured in our data. The “exposures-per-worker” metric serves as an indicator that correlates to the potential for exposure to carcinogens in a given sector or occupation. It should be used and interpreted with caution, as workers may be exposed to multiple agents and thus counted multiple times in their respective sector or occupation [1]. Additionally, with current data limitations, it is impossible to account for additive effects of included agent exposures or other lifestyle factors (such as smoking) that could elevate a worker’s risk of cancer [1].

There was a disparity in sampled populations between the Prairies and Ontario, with the Prairies having a higher sampled population despite a smaller overall population, which is attributed to survey methodology intricacies. Ontario witnessed a substantial decrease of over 900 sampled households per month, offset by increased sample sizes in the three Prairie Provinces and Quebec [14]. Conversely, alterations in sample sizes for the Atlantic Provinces and British Columbia were comparatively modest [14].

Young age was a significant predictor of being a new worker, and this has been consistently demonstrated in Canada and other countries [20,21]. This is an important finding from an occupational health and safety standpoint, as younger workers are at an elevated risk of job-related injuries and potential exposure to carcinogens and other hazards [22,23,24,25,26]. Previous research from our team and others has demonstrated an added potential exposure to hazardous occupational environments [21,22,23,25,26,27,28]. Numerous studies dedicated to occupational safety across Canada and beyond consistently underscore the heightened risk to young and entry-level workers of more frequent occupational injuries as opposed to their more seasoned counterparts [22,27,28]. Additionally, these injuries and the associated hazards are often linked at least partly to limited safety awareness and poor compliance with proper safety protocols [24,28].

The association with younger age is particularly concerning given the level of carcinogenic exposure in many of the occupations where new workers are overrepresented. New workers are often younger and have a well-established increased risk of occupational injury [29], which may extend to an increased risk of exposure to carcinogens [13]. Factors include biological elements (e.g., incomplete cognitive development until around age 25), limited jobsite experience, lack of understanding of hazardous settings, perceived lower ability to voice safety concerns, and a lack of training [13,14,15,16]. Many of these factors that increase workplace risk among younger workers can be applied to new workers generally and suggest that new workers are more vulnerable to carcinogenic exposures than those in more permanent employment.

We found that new workers are exposed to several hazards deemed either carcinogenic or probably carcinogenic by IARC, notably solar radiation, pesticides (glyphosate), wood dust, and silica [30]. These substances are associated with increased risk of lung, laryngeal, nasal cavity, and skin cancer and lymphoma (Hodgkin’s and non-Hodgkin’s), as well as silicosis, pulmonary tuberculosis, chronic obstructive pulmonary disease, and cataracts, among others [30]. As many of these diseases can have lifelong health impacts, mitigating exposure should be a priority concern for young workers in industries with the highest exposures. These exposures also often occur in occupation groups where employees are precariously employed [31], specifically in natural resources and agriculture, where there is also the highest proportion of new workers (20.5% versus 11.0% nationally). It is also important to note that, although asbestos was not included in our analysis, as it is not a common exposure, some workers may still be at risk of exposure [32]. Asbestos was banned in Canada in 2018 but older homes and buildings may still contain insulation made with asbestos [33]. This places construction workers at risk of exposure who may be involved in the remediation or demolition of older buildings, putting them at risk of mesothelioma and lung cancer [32]. Preventing asbestos exposure among workers in these occupations should continue to be a priority.

New workers were also more likely to be new immigrants, unlike their Canadian counterparts in our study, which identifies another priority population for possible intervention. Newcomers to Canada face unique challenges in the workforce, particularly concerning working conditions and occupational health [34]. Recent immigrants are likely to face challenges securing employment in their preferred profession, resulting in low-paying and potentially more hazardous jobs [35,36]. Additionally, new immigrants may lack awareness of Canadian labour laws, including the right to refuse unsafe work, and resources for work-related injury support [35,37].

The highest exposures-per-work metric occurred among equipment operators in the trades and transport occupational groups and a modest overrepresentation of new workers compared to the workforce as a whole, who may have frequent exposure to solar radiation and engine emissions. Epidemiologic studies have extensively investigated the effects of particulate matter emitted from gasoline and diesel engines, especially in occupational settings, and classified diesel engine emissions as carcinogenic to humans [38,39,40,41,42]. Exhaust fumes have been linked to several adverse health effects, including renal cancer [43], laryngeal cancer [44], and lung cancer [45]. In Ontario alone, over 200 new cases of lung cancer annually are linked to workers’ exposure to diesel exhaust [46], of which a third (33%) are motor transport operators [46]. These findings highlight the importance of implementing measures to reduce diesel emissions and protect the health of workers.

Several key strengths and limitations to this study should be highlighted. The LFS is a large, recurring, population-based survey which makes the findings generally representative of the working population in Canada. One year (2017) of data was used for this study, selected because it was the most recent public use microdata file available that collected information on immigration status. The 12 monthly surveys from 2017 were pooled to examine the impact of season on new worker status, which is why no survey weights were applied to the analysis. This approach may have double-counted a small number of workers who were asked to fill in the survey more than once that year. The prevalence of new workers was compared by industry and occupation with a semi-quantitative indicator of hazardous exposures from the CAREX system [19]. CAREX Canada is a longstanding, large, and reliable program of research on occupational exposure to carcinogens in Canada, which is a strength of this approach. However, this comparison only serves as an indicator of potentially hazardous working environments for new workers and does not allow comment for any individual or group exposed to carcinogens and severity. Since exposure monitoring data are not available by new worker status, age, or any other demographic dimension, it was required to consider potentially hazardous exposures using the CAREX system. CAREX Canada also only considers exposures to known and suspected carcinogens, and there are many hazards in the industries and occupations identified in our qualitative comparison of new workers as a proportion of the worker group with exposure metrics. Even if the exposure-per-worker metric was low for certain sectors and occupations in our findings (e.g., sales and service occupations), this does not mean these jobs do not carry excess risk of occupational disease for the new workers who make up a large proportion of these worker groups.

Another limitation to the analysis is the industry and occupation data in the LFS, which were based solely on the respondents’ main job the week before the survey was taken. The new worker respondents may likely have held part-time and/or temporary work which may not have been captured in the analysis and could have affected the analysis. Finally, the definition of ‘new worker’ used in this study only takes into account the length of an employee’s tenure and may miss details on those who are changing roles within the same organization.

## 5. Conclusions

In conclusion, this study has shed light on predictors of being a new worker and explores how these jobs may also experience more hazardous exposures that place them at risk of occupational cancer. A disproportionate number of new workers are employed in industries and occupations (largely related to agriculture and construction) where there are also myriad exposures that can cause occupational cancer. There is a stark lack of data on occupational exposures that take into account demographic factors like age, immigration status, and job tenure, which is why only inferences about the hazard profile experienced by new workers can be made. Employers should ensure that new workers, particularly young and immigrant workers, receive adequate training prior to starting a job. Training will not replace systemic protections for workers but may help ease the immediate burden of injury and illness. This study highlighted an important gap in the understanding of the occupational health risks faced by new workers.

A critical next step in addressing the gap in knowledge of new workers is to collect data on occupational exposures for new workers in specific industries and occupations. This is the only way that researchers will be able to assess whether there are differences in exposure for this population compared to other workers, and this study is a call to action to address this data gap. Further research in this area is crucial to advance the understanding of the diverse experiences of new workers and promote inclusive occupational disease prevention.

## Figures and Tables

**Figure 1 ijerph-21-01013-f001:**
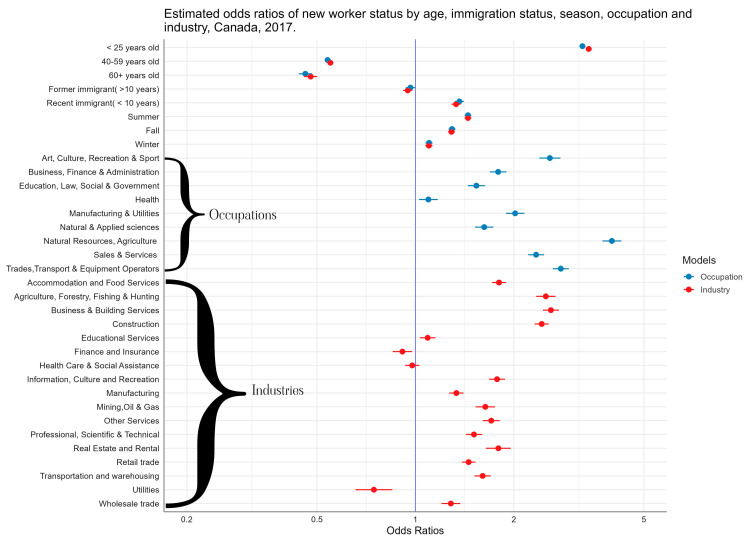
Estimated odds ratios of new worker status by age, immigration status, season, occupation, and industry, Canada LFS, 2017.

**Table 1 ijerph-21-01013-t001:** Bivariate relationships between demographic characteristics and new worker status in the 2017 Labour Force Surveys.

		New Worker Status		
	Total Workforce	Yes ^1^	No ^1^	
Characteristic	N = 608,724	66,775 (11.0%)	541,949 (89.0%)	*p* ^2^
**Age (years)**				<0.001
<25	91,715 (15.1)	27,512 (41.2)	64,203 (11.8)	
25–39	204,165 (33.5)	21,422 (32.1)	182,743 (33.7)	
40–59	271,763 (44.6)	15,730 (23.6)	256,033 (47.2)	
60+	41,081 (6.7)	2111 (3.2)	38,970 (7.2)	
**Sex**				<0.001
Female	306,938 (50.4)	31,254 (46.8)	275,684 (50.9)	
Male	301,786 (49.6)	35,521 (53.2)	266,265 (49.1)	
**Education Attainment**				<0.001
0–8 years	8417 (1.4)	1269 (1.9)	7148 (1.3)	
Some high school	49,521 (8.1)	10,113 (15.1)	39,408 (7.3)	
High school graduate	124,480 (20.4)	15,135 (22.7)	109,345 (20.2)	
Some postsecondary	41,602 (6.8)	7705 (11.5)	33,897 (6.3)	
Postsecondary/Diploma	230,205 (37.8)	20,791 (31.1)	209,414 (38.6)	
Bachelor’s degree	110,010 (18.1)	8665 (13.0)	101,345 (18.7)	
Above Bachelor	44,489 (7.3)	3097 (4.6)	41,392 (7.6)	
**Immigration status**				<0.001
Citizen	511,069 (84.0)	56,753 (85.0)	454,316 (83.8)	
Former immigrant (>10 years)	60,275 (9.9)	4666 (7.0)	55,609 (10.3)	
Recent immigrant (<10 years)	37,380 (6.1)	5356 (8.0)	32,024 (5.9)	
**Industry (NAICS)**				<0.001
Accommodation and Food Services	44,403 (7.3)	8950 (13.4)	35,453 (6.5)	
Agriculture, Forestry, Fishing, and Hunting	9562 (1.6)	1761 (2.6)	7801 (1.4)	
Business and Building Services	19,848 (3.3)	3487 (5.2)	16,361 (3.0)	
Construction	42,583 (7.0)	7078 (10.6)	35,505 (6.6)	
Educational Services	50,543 (8.3)	3408 (5.1)	47,135 (8.7)	
Finance and insurance	24,129 (4.0)	1451 (2.2)	22,678 (4.2)	
Health Care and Social Assistance	87,596 (14.4)	5806 (8.7)	81,790 (15.1)	
Information, Culture and Recreation	22,226 (3.7)	3527 (5.3)	18,699 (3.5)	
Manufacturing	62,840 (10.3)	5503 (8.2)	57,337 (10.6)	
Mining, Oil, and Gas	15,239 (2.5)	1531 (2.3)	13,708 (2.5)	
Other services	21,461 (3.5)	2675 (4.0)	18,786 (3.5)	
Professional, Scientific, and Technical	28,444 (4.7)	2895 (4.3)	25,549 (4.7)	
Public Administration	40,465 (6.6)	2432 (3.6)	38,033 (7.0)	
Real estate and rental	6896 (1.1)	783 (1.2)	6113 (1.1)	
Retail trade	77,049 (12.7)	10,742 (16.1)	66,307 (12.2)	
Transportation and warehousing	29,114 (4.8)	2807 (4.2)	26,307 (4.9)	
Utilities	5874 (1.0)	271 (0.4)	5603 (1.0)	
Wholesale trade	20,452 (3.4)	1668 (2.5)	18,784 (3.5)	
**Occupation (NOC)**				<0.001
Manufacturing and Utilities	33,087 (5.4)	3289 (4.9)	29,798 (5.5)	
Art, Culture, Rec, and Sport	10,699 (1.8)	1960 (2.9)	8739 (1.6)	
Business, Finance, and Administration	97,202 (16.0)	8006 (12.0)	89,196 (16.5)	
Education, Law, Social, and Government	72,551 (11.9)	5217 (7.8)	67,334 (12.4)	
Health	50,006 (8.2)	2769 (4.1)	47,237 (8.7)	
Management	35,902 (5.9)	1427 (2.1)	34,475 (6.4)	
Natural Resources, Agriculture	16,349 (2.7)	3357 (5.0)	12,992 (2.4)	
Natural and Applied Science	42,053 (6.9)	3286 (4.9)	38,767 (7.2)	
Sales and Service	157,631 (25.9)	24,785 (37.1)	132,846 (24.5)	
Trades, Transport and Equipment Operators	93,244 (15.3)	12,679 (19.0)	80,565 (14.9)	
**Union membership**				<0.001
Contract	11,771 (1.9)	1461 (2.2)	10,310 (1.9)	
Member	188,665 (31.0)	9964 (14.9)	178,701 (33.0)	
Non-unionized	408,288 (67.1)	55,350 (82.9)	352,938 (65.1)	
**Firm size (employees)**				<0.001
<20	115,292 (18.9)	18,796 (28.1)	96,496 (17.8)	
20–99	101,250 (16.6)	13,665 (20.5)	87,585 (16.2)	
100–500	89,964 (14.8)	9505 (14.2)	80,459 (14.8)	
500+	302,218 (49.6)	24,809 (37.2)	277,409 (51.2)	
**Job type**				<0.001
Contract	41,380 (6.8)	12,667 (19.0)	28,713 (5.3)	
Temporary	21,414 (3.5)	5782 (8.7)	15,632 (2.9)	
Permanent	525,539 (86.3)	40,164 (60.1)	485,375 (89.6)	
Seasonal	20,391 (3.3)	8162 (12.2)	12,229 (2.3)	
**Hourly rate**				<0.001
CAD 15.00 and less	150,664 (24.8)	33,344 (49.9)	117,320 (21.6)	
CAD 15.25–CAD 22.00	152,057 (25.0)	16,436 (24.6)	135,621 (25.0)	
CAD 22.00–CAD 32.50	152,576 (25.1)	10,223 (15.3)	142,353 (26.3)	
> CAD 32.50	153,427 (25.2)	6772 (10.1)	146,655 (27.1)	
**Season**				<0.001
Spring	150,995 (24.8)	13,964 (20.9)	137,031 (25.3)	
Summer	155,501 (25.5)	20,540 (30.8)	134,961 (24.9)	
Fall	152,760 (25.1)	17,480 (26.2)	135,280 (25.0)	
Winter	149,468 (24.6)	14,791 (22.2)	134,677 (24.9)	
**Region**				<0.001
Atlantic	92,331 (15.2)	10,822 (16.2)	81,509 (15.0)	
British Columbia	70,410 (11.6)	8591 (12.9)	61,819 (11.4)	
Ontario	163,118 (26.8)	16,780 (25.1)	146,338 (27.0)	
Prairies	176,954 (29.1)	19,177 (28.7)	157,777 (29.1)	
Quebec	105,911 (17.4)	11,405 (17.1)	94,506 (17.4)	

^1^ n (%); ^2^ Pearson’s Chi-squared test; *p*: adjusted *p*-value using Bonferroni correction for multiple testing; NAICS: North American Industry Classification System Canada 2017 Version 1.0; NOC: National Occupational Classification 2016 Version 1.0.

**Table 2 ijerph-21-01013-t002:** Proportion of new workers by sector (industry) and CAREX Canada exposure prevalence data, 2016.

Industry	New Workers (%)	Common Exposures	Exposures-per-Worker Metric
Accommodation and Food Services	20.2	Night shift work, Gasoline engine exhaust, Polycyclic aromatic hydrocarbons	0.44
Agriculture, Forestry, Fishing, and hunting	18.4	Solar radiation, Glyphosate, 2,4-D	1.29
Business, building & other support services	17.6	Solar radiation, Gasoline engine exhaust, Night shift work	0.56
Construction	16.6	Solar radiation, Silica, Wood dust	1.34
Information, culture and Recreation	15.9	Night shift work, Solar radiation, Gasoline engine exhaust,	0.21
Retail trade	13.9	Night shift work, Gasoline engine exhaust, Polycyclic aromatic hydrocarbons	0.28
Other services	12.5	Gasoline engine exhaust, Polycyclic aromatic hydrocarbons, Welding fumes	0.63
Real estate and rental	11.4	Gasoline engine exhaust, Night shift work, Solar radiation,	0.43
Professional, Science and Tech	10.2	Solar radiation, Gasoline engine exhaust, Ionizing radiation	0.14
Mining, quarrying, Oil and Gas	10.0	Diesel engine exhaust, Solar radiation, Gasoline engine exhaust	1.18
Transportation and warehousing	9.6	Diesel engine exhaust, Gasoline engine exhaust, Solar radiation	1.61
Overall workforce	11.0	Night shift work, Solar radiation, Gasoline engine exhaust	0.56

**Table 3 ijerph-21-01013-t003:** Proportion of new workers by occupation and CAREX Canada exposure prevalence data, 2016.

Occupational Groups	New Workers (%)	Common Exposures	Exposures-per-Worker Metric
Natural Resources, Agriculture and related production	20.5	Solar radiation, Gasoline engine exhaust, Diesel engine exhaust	1.57
Art, culture, recreation and sport	18.3	Solar radiation, Night shift work, Chloroform	0.19
Sales and service occupations	15.7	Polycyclic aromatic hydrocarbons, Gasoline engine exhaust, Night shift work	0.35
Trades, transport and equipment operators	13.6	Gasoline engine exhaust, Diesel engine exhaust, Solar radiation	1.98
Manufacturing and utilities	9.9	Night shift work, Benzene, Wood dust	0.77
Business, Finance, and Administration	8.2	Night shift work, Gasoline engine exhaust, Solar radiation	0.14
Natural and applied sciences and related occupations	7.8	Solar radiation, Gasoline engine exhaust, Night shift work	0.20
Education, law and social, community and government services	7.2	Solar radiation, Gasoline engine exhaust, Night shift work	0.23
Health care	5.5	Night shift work, Antineoplastic agents, Ionizing radiation	0.38
Management	4.0	Solar radiation, Night shift work, Second hand smoke	0.15
Overall workforce	11.0	Night shift work, Solar radiation, Gasoline engine exhaust	0.56

## Data Availability

The data are hosted by the Abacus Data Network Canada and are available in the public domain at https://hdl.handle.net/11272.1/AB2/PO8CWI (accessed on 31 July 2024).

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
