# Peer review of "Overrepresentation of New Workers in Jobs with Multiple Carcinogen Exposures in Canada"

_ijerph, 2024, doi:10.3390/ijerph21081013_

Round 1

Reviewer 1 Report

Comments and Suggestions for Authors

Thank you very much for inviting me to review this manuscript. The manuscript presents sociodemographic and job-related factors among new workers and the exposure metrics related to carcinogens in each industry using the 2017 Labor Force Survey.

Given that this is a cross-sectional survey, the scientific value of the first part may be limited. However, the second part of the study incorporates data from CAREX Canada to understand the potential exposure metrics, which has the potential to add significant value to the literature. Nonetheless, the interpretation of the exposures-per-worker metric is unclear. For example, the overall workforce has an exposures-per-worker metric of 0.56 (Table 2 and Table 3), but this value is not easily understood. This issue also applies to each occupation and industry in Table 2 and Table 3. It would be beneficial to clearly explain the exposures-per-worker metric in the Methods and provide guidance on how it should be interpreted in Results and Discussion sections.

Given the ambiguity of the exposures-per-worker metric and the proportion data presented in Table 2 and Table 3, it may be valuable to show the burden, i.e., the number of new workers exposed to each type of carcinogen annually, as well as to each occupation or industry. These estimated numbers could significantly contribute to policy applications.

In addition to describing the sociodemographic and job-related factors among new workers, the study could contribute more to the literature by analyzing factors related to the total burden of cancer exposure.

Comments on the Quality of English Language

The flow of English is good. However, the manuscript format should be carefully checked, including table formatting, citations, and the cropped text in Figure S1.

Author Response

Comment 1: Nonetheless, the interpretation of the exposures-per-worker metric is unclear. For example, the overall workforce has an exposures-per-worker metric of 0.56 (Table 2 and Table 3), but this value is not easily understood. This issue also applies to each occupation and industry in Table 2 and Table 3. It would be beneficial to clearly explain the exposures-per-worker metric in the Methods and provide guidance on how it should be interpreted in Results and Discussion sections.

Thank you for this comment. We agree that the interpretation of “exposures-per-worker” can be challenging and we have added an explanation to the Methods and Discussion sections.

Line 123-125: This metric can be interpreted as a measure of the presence of carcinogens in an occupation when there are expected to be multiple carcinogenic exposures.

Line 236-242: The "exposure per worker" metric serves as an indicator that correlates to the level of severity of exposure to carcinogens in a given sector or occupation. It should be used and interpreted with caution, as workers may be exposed to multiple agents and thus counted multiple times in their respective sector or occupation1. Additionally, with current data limitations, it is impossible to account for additive effects of included agent exposures or other lifestyle factors (such as smoking) that could elevate a worker’s risk of cancer1.

Comment 2: Given the ambiguity of the exposures-per-worker metric and the proportion data presented in Table 2 and Table 3, it may be valuable to show the burden, i.e., the number of new workers exposed to each type of carcinogen annually, as well as to each occupation or industry.

While we agree that it would be beneficial to have the number of new workers exposed to carcinogens, this is not possible with the data available to us. As we mentioned in the methods, there are no data on occupational exposures for new workers measured in workplaces to examine differences in exposure levels by job tenure. Therefore, our estimates of exposure risk are the most we can do given the current data. In Table 1 we do provide the number of new workers in occupations and industries and we hope that more data will be available in the future to adequately assess the burden. As suggested, we have added a call for additional data to be collected among new workers in the discussion.

Lines 345-349: A critical next step in addressing the gap in knowledge on new workers is to collect data on occupational exposures for new workers in specific industries and occupations. Using these data, we may be able to assess whether there are differences in exposure for this population compared to other workers.

Reviewer 2 Report

Comments and Suggestions for Authors

This article presents a particularly original and important analysis of the overexposure to carcinogens of workers recently recruited to an exposed job, highlighting the fact that the majority of these workers are young, migrant or even seasonal workers, many of them unskilled.

The article is clear in its presentation of objectives and methodology, as well as in the presentation and discussion of results.

The identification of numerous situations of co-exposure to several carcinogens further exacerbates the effect of the very significant overexposure of the workers concerned.

These results confirm a very worrying situation, already documented in the European Union https://osha.europa.eu/en/facts-and-figures/workers-exposure-survey-cancer-risk-factors-europe , and in France in particular through DARES surveys https://dares.travail-emploi.gouv.fr/publication/les-expositions-des-salaries-aux-produits-chimiques-cancerogenes#:~:text=Les%20salariés%20des%20petits%20établissements,que%20dans%20les%20grands%20établissements .

It might be useful to mention these references in the bibliography.Among the carcinogens mentioned as being the most frequent in the construction sector, asbestos is not mentioned, even though it is an ever-present risk factor, given the extent of asbestos use and the absence of asbestos removal in many buildings since the bans.  It might be useful to indicate why this major carcinogen is not included among the substances selected, at least in the discussion.

Author Response

Comment 1: It might be useful to mention these references in the bibliography.

Thank you for providing these resources. We have reviewed them and added the French report to the manuscript at line 256 and the European findings to the manuscript on line 263.

Comment 2: Among the carcinogens mentioned as being the most frequent in the construction sector, asbestos is not mentioned, even though it is an ever-present risk factor, given the extent of asbestos use and the absence of asbestos removal in many buildings since the bans.  

Thank you for bringing this to our attention. Asbestos was not originally included in our analysis because we only analyzed the most common exposures to workers. However, we agree that it is important to include given that asbestos was banned several years ago. We have added a point in the discussion indicating that some workers are still at risk of exposure to asbestos, especially those in the construction industry.

Lines 285-292: It is also important to note that, although asbestos was not included in our analysis as it is not a common exposure, some workers may still be at risk at risk of exposure. Asbestos was banned in Canada in 2018 but older homes and buildings may still contain insulation made with asbestos. This places construction workers at risk or exposure who may be involved in the remediation or demolition of older buildings, putting them at risk of mesothelioma and lung cancer. Preventing asbestos exposure among workers in these occupations should continue to be a priority.

We have also reformatted the figure where text was cut off and we have proofread the manuscript in full. We recognize that there are some unformatted citations that will be changed prior to publication.

Round 2

Reviewer 1 Report

Comments and Suggestions for Authors

Thank you to the authors for their response. There is one comment that remains unaddressed:

In addition to describing the sociodemographic and job-related factors among new workers, the study could contribute more to the literature by analyzing factors related to the total burden of cancer exposure.

Author Response

Comment 1: “In addition to describing the sociodemographic and job-related factors among new workers, the study could contribute more to the literature by analyzing factors related to the total burden of cancer exposure.”

Response : We must reiterate our response to a previous similar comment – we completely agree that assessing the burden of carcinogen exposure among new workers is a critical gap.

Lines 286-288 : We simply do not have the data that would enable any analysis or calculation of burden of exposure in this group. Our manuscript is meant as a call to action to address this gap. We have adjusted our previous revision’s final paragraph to be clearer about this fact.